# Patient Experience of Flunarizine for Vestibular Migraine: Single Centre Observational Study

**DOI:** 10.3390/brainsci12040415

**Published:** 2022-03-22

**Authors:** Sk Mamun Ur Rashid, Sheetal Sumaria, Nehzat Koohi, Qadeer Arshad, Diego Kaski

**Affiliations:** 1Department of Neuro-Otology, Royal National Ear Nose and Throat Hospital, University College London Hospitals, London WC1E 6DG, UK; sk.rashid@nhs.net; 2Department of Pharmacy, National Hospital for Neurology and Neurosurgery, London WC1N 3BG, UK; s.sumaria@nhs.net; 3Department of Clinical and Movement Neurosciences, University College London, London WC1N 3BG, UK; n.koohi@ucl.ac.uk (N.K.); qa15@leicester.ac.uk (Q.A.); 4InAmind Laboratory, Department of Psychology, Neuroscience and Behaviour, University of Leicester, Leicester LE1 7RH, UK

**Keywords:** vestibular migraine, flunarizine, prophylaxis, symptoms, patient experience

## Abstract

Vestibular migraine (VM) is a leading cause of episodic vertigo, affecting up to 1% of the general population. Despite established diagnostic criteria, there is currently no evidence-based approach for acute treatment of VM, with treatment recommendations generally extrapolated from studies on classical migraine headache. Several small-scale studies have identified flunarizine as a potentially effective prophylactic medication in VM. We conducted a single-centre observational service evaluation study exploring patient experiences of preventative medications over a 28-month period, including flunarizine, for control of VM symptoms. To compare patient experience of flunarizine with other medications, data from patients taking flunarizine were separately analysed. A total of 90% of VM patients taking flunarizine reported symptomatic improvement, compared to only 32% of patients on other medications. Whilst 50% of patients on flunarizine reported side effects. these were not deemed to outweigh the clinical benefits, with most patients deciding to continue treatment. Our data supports the use of flunarizine in VM.

## 1. Introduction

Vestibular migraine (VM) is a leading cause of episodic vertigo, affecting up to 1% of the general population and 11% of patients in specialized dizziness clinics [1]. The development of diagnostic criteria by the Committee for Classification of Vestibular Disorders of the Bárány Society [2] has engendered a broader acceptance of VM within neurological communities and was an important step in bringing together diagnostic standards for this condition.

Despite established diagnostic criteria, there is currently no evidence-based approach for acute treatment of VM, with treatment recommendations generally extrapolated from studies on classical migraine headache. Prophylactic medication for VM is appropriate if attacks are frequent and/or insufficiently controlled by non-pharmacological measures (e.g., lifestyle measures).

A recent systematic review and meta-analysis assessing the efficacy of preventative treatments for VM identified that antiepileptic drugs, calcium channel blockers, tricyclic antidepressants, beta-blockers, serotonin, and norepinephrine reuptake inhibitors, as well as vestibular rehabilitation improve outcome parameters [3]. However, due to significant heterogeneity of studies and lack of standardised reporting outcomes, a preferred treatment modality could not be determined. A prospective multi-centre study evaluating acetazolamide, amitriptyline, flunarizine, propranolol or topiramate for VM, found similar reduction in symptom severity and frequency across all drugs [4]. A further prospective randomized non-placebo-controlled trial suggested flunarizine is effective in reducing the severity and frequency of vertigo attacks in VM patients [5].

Here, we report subjective outcomes in patients taking flunarizine for VM and compare these outcomes to patients taking other migraine prophylactic agents in the same clinic.

## 2. Materials and Methods

This was a retrospective, single-centre observational study of 72 patients with definite or probable vestibular migraine, according to the International Classification of Headache Disorders -3 (ICHD-3) criteria [6]. Patient details were retrieved on the electronic patient record system. Only patients taking a single prophylactic medication at the time of interview were included. Non-treating team members then conducted a semi-structured telephone interview to explore patient experiences on prophylactic medications, focusing on current dose, change in symptoms since starting medication, side effects, and use of previous migraine prophylactic medication. Patients were specifically asked to state whether they had found the preventative medication(s) to reduce the severity and frequency of their vestibular attacks (yes/no; primary outcome). Patients were divided into groups according to prophylactic medication, Group 1 receiving ‘flunarizine’, and Group 2 ‘other medication’. Flunarizine has been available to prescribe for VM in our Centre since 2017, with advice to do so in patients who have not responded to at least 1 other preventative medication.

## 3. Results

Among the 72 patients approached, 56 were contactable, completed the telephone interview and are included in the descriptive analysis. Of these, 47 were female and 9 males. Mean age was 47 years (range 24 to 70 years). Medications were prescribed to patients between June 2019 to October 2021 by the same clinician. Furthermore, 28 patients were allocated to the ‘flunarizine group’ and 28 to the ‘other medication’ group (Table 1). There were no gender, age, or disease duration differences between medication groups (*p* > 0.05; Table 1). Two of the patients in the flunarizine group did not start the medication although this was prescribed. All the patients had been on flunarizine for a minimum of three months. In the flunarizine group patients started with 5 mg daily dosage, apart from 2 patients who started on 10 mg daily. In the flunarizine group, 21 patients remained on a 5 mg dose whereas three patients subsequently increased the dose to 10 mg due to ongoing symptoms; 13 patients in the flunarizine group had >15 days of dizziness or vertigo per month and the remainder at least once weekly before starting flunarizine. 25 participants reported overall symptomatic improvements with flunarizine (90%), with only 3 patients reporting no changes in symptoms (Figure 1). Patients taking flunarizine reported improvements in both duration (*n* = 22, 78%) and severity (*n* = 24, 85%) of their symptoms. A majority (*n* = 26, 92%) of these patients had already tried previous preventative medications before taking flunarizine (Figure 2), as per prescription guidelines in our Centre. Moreover, 13 (50%) patients experienced side-effects from flunarizine that included weight gain (*n* = 4, 31%), sleep disturbance and mood changes, but only 1 patient discontinued the medication because of side effects (weight gain). In the ‘other medications’ group, 8 patients were taking amitriptyline (40%) and seven were taking candesartan (25%), six were taking nortriptyline (21%), and the remainder sertraline (*n* = 1, 3%), gabapentin (*n* = 1, 3%), or topiramate (*n* = 2, 7%). All of patients had been on these medications for a minimum of three months. In this group, 71% (*n* = 20) of patients had already tried previous preventive medications, and three patients had previously also undergone a 6-week period of vestibular rehabilitation. In the ‘other medication’ group, 32% (*n* = 9) reported symptomatic improvement with the medication (Figure 1), mostly in relation to severity (*n* = 7) rather than frequency of attacks (*n* = 4). A total of 21 (75%) patients experienced side-effects from preventative medications in this group that included weight gain (40%), sleep disturbance (50%), palpitations (20%) and mood changes (40%), and hives (*n* = 1).

## 4. Discussion

The intended benefit of prophylactic treatment for VM is to reduce severity, duration, and frequency of vertigo attacks. Here, we explored patient experiences on flunarizine as compared to other VM prophylactic medications. Flunarizine is a calcium channel blocker which was found to be effective in the prophylaxis of migraine headache and vertigo treatment [7], and our data corroborate this finding. We show that patient experience with flunarizine is substantially superior to alternative migraine prophylactic agents in VM. Frequency of side effects was similar across the flunarizine and ‘other treatment’ groups, but only one patient taking flunarizine discontinued its use due to side effects. Our findings are particularly intriguing given that both groups were matched for age, gender, disease duration, and disease ‘activity’ (average number of vertigo attacks per month).

Lepcha and colleagues [8] conducted an unblinded, controlled randomised trial where 52 patients were given ‘as needed’ paracetamol and betahistine and instructed to perform vestibular exercises, with half of the patients also given flunarizine 10 mg daily for 12 weeks. In the post-intervention analysis, 88% of patients receiving flunarizine compared to 52% of patients not receiving flunarizine reported having ‘low vertigo frequency’ (2–3 attacks or less per 3 months) as per a unique 6-point Likert scale; 88% of patients reported a ‘marked improvement’ with flunarizine compared to 61% without flunarizine. However, the study did not report outcome measures at baseline, and multiple scale points were grouped together arbitrarily to determine ‘low frequency’ and ‘marked improvement’. There were no significant differences in headache measures between the two groups. Thus, our results are consistent with the levels of symptomatic improvement of vestibular symptoms in patients taking flunarizine. Wouters et al. also found that either loading dose or fixed dosage flunarizine improves dizziness symptoms and was well-tolerated [9]. In a retrospective VM study Baier et al. reported symptomatic improvements of episodic vertigo attacks with flunarizine compared to a no treatment arm [10].

One retrospective study compared different medications which were prescribed based on clinician discretion [4]. Patients were treated with amitriptyline 25–50 mg (*n* = 15), flunarizine 10 mg (*n* = 11), propranolol 40–80 mg (*n* = 7), topiramate 100–200 mg (*n* = 8), sodium valproate 500–1000 mg (*n* = 3), nortriptyline 50 mg (*n* = 1) or venlafaxine 75 mg (*n* = 1) for 3 months. Complete data were not available for all patients and a before-and-after analysis was performed. When all of the patients were analysed, there was a significant improvement in all outcome measures (*p* < 0.001). The authors found significant improvements in self-reported state (amitriptyline, flunarizine and propranolol 80 mg), headache severity as measured by VSS (amitriptyline 25 mg, flunarizine 10 mg, propranolol 80 mg and topiramate 100 mg) and vestibular severity as measured by vertigo symptom scale (VSS [amitriptyline 25 mg, flunarizine 10 mg and propranolol 80 mg]). There were no between-group differences for any of the outcome measures but the study was not sufficiently powered to detect such differences [4]. Liu et al., evaluated the efficacy and safety of venlafaxine, flunarizine, and valproate in a randomized comparison trial for VM prophylaxis [11]. All of the medications resulted in reduced dizziness handicap inventory (DHI) scores, without between-group differences, but flunarizine and venlafaxine also reduced the VSS scores, which was not observed with valproate. Our study was explicitly designed to explore patient experiences with these medications, which one might expect to correlate with quantitative questionnaire scores of symptom severity. Such quantitative measures are, however, best explored through randomized controlled trials.

Dominguez-Duran and colleagues conducted a prospective observational study which compared five different treatments: acetazolamide 250 mg, amitriptyline 10 mg, flunarizine 5 mg, propranolol 10 mg and topiramate 25 mg, all given daily for 5 weeks [12]. The doses were lower and duration of treatment shorter than in other studies. A total of 50 patients were recruited, eight took medication less than 20% of the time and 11 were lost to follow up, leaving 31 in the final analysis. The majority of 31 patients analysed were prescribed amitriptyline (*n* = 16), with 5 receiving topiramate and acetazolamide, 4 propranolol and only 1 flunarizine; individual outcomes for the latter were not reported.

### 4.1. Side Effects

Across studies, calcium channel blockers have a higher rate of reported side effects than other agents, such as beta-blockers, with 75% of patients discontinuing flunarizine in the Dominiguez-Duran et al. study (which was the greatest drop out of all the treatment arms) [12]; in the Lepcha et al. study, 24% of patients experienced side effects compared with 9% in the control group [8]; and, Liu et al. reported that five (of 22) patients experienced side effects on flunarizine (compared with two patients each in the valproate and venlafaxine groups) [11]. This was not our experience, where similar rates of side effects were observed across all medications used, and only a single patient discontinued flunarizine due to side effects.

### 4.2. VM Pathophysiology

The pathophysiology of VM is incompletely understood [13], with contributions from environmental [14], endogenous (hormonal) [15], and genetic factors [13,14,16,17]. As with migraine, there is a significant female preponderance [18,19] for reasons not well explained, but reflected in our data set also with over 80% of our cohort being female. There are two major putative mechanisms to account for acute attacks: (1) hypoperfusion of the inner ear during migrainous attacks secondary to vasospasm resulting in vertiginous symptoms. This is supported by the association of migraine with sudden sensorineural hearing loss [20] and the observation that migraine as a risk factor for cerebrovascular accidents [21]; (2) sensitisation and activation of the trigeminovascular (TV) system leading to release of the pro-inflammatory neuropeptides substance *p* and calcitonin gene-related peptide (CGRP), which has connections with brain areas associated with processing of nociceptive information as well as thalamic and vestibular-associated cortices [22]. Neuroimaging studies support the hypothesis that there are specific abnormalities in the structure and activity of the vestibulo-thalamo-cortical pathway in vestibular migraine [23,24], although other structures, such as the insular cortex, may also be involved [25].

Cortical spreading depression (CSD)’ and ‘ion channel defect (ICD)’ hypotheses have also been put forward to explain VM pathophysiology more generally (9). In classical migraine, during CSD neocortical and extracellular release of ions, namely K^+^, H^+^, Ca^2+^, nitric oxide arachidonic acid and calcitonin gene-related peptide (CGRP), activate a TV reflex mediated vasodilation of the meningeal vessels resulting in pain perception by activation of ascending thalamocortical pathways [26,27]. The TV reflex system also innervates inner ear blood supply causing rapid vasodilatation, possibly accounting for vestibular symptoms in VM through activation of peripheral and central afferent pathways between the vestibular sensory organs and the medulla and pons [26]. Cav2.1 voltage dependent calcium channels were shown to be mediators of the TV reflex which, in addition, regulate CGRP release from neuronal processes in the dura, trigeminal ganglion, the spinal trigeminal nucleus, and the inner ear. At the vestibular receptor level, flunarizine may prevent influx of calcium ions into the cell reducing the magnitude of mechano-electrical transduction which is mainly dependent on the endolymphatic calcium concentration. Flunarizine may also attenuate changes induced by CSD in the brain by reducing the damage caused by oxidative stress and mitochondrial injury [28].

Another putative mechanism that could account for the efficacy of flunarizine in VM is a dopamine dysfunction hypothesis, given the known effects of flunarizine upon dopamine regulation [29]. The modulation of the dopaminergic system has been associated with the pathophysiology of migraine [30,31]. This linkage is corroborated by two observations. Firstly, migraineurs compared to controls are hypersensitive to administration of dopamine agonists (i.e., eliciting vegetative symptoms of nausea and yawning) [32]. Secondly, symptoms of acute migraine can be controlled via dopamine receptor antagonists [33].

### 4.3. Use of Flunarizine in the UK

Whilst being widespread throughout Europe and the USA, the use flunarizine is not licensed in the UK for migraine prophylaxis. National Health Service prescription cost analysis for England in 2013 reported that flunarizine hydrochloride 5 mg capsules cost £115.42 per item. A Cochrane review in 2015 was unable to make any recommendations for pharmacological prevention in VM due to lack of data but instead suggested adoption of strategies with proven efficacy in classical migraine treatment [34]. In numerous studies flunarizine was shown to be effective against vertiginous attacks with very few medium-term adverse effects compared to commonly used medications for migraine, and this is supported by our data. 

Caution must be adopted when interpreting our findings however as this was a retrospective review of patient symptoms at a single centre. We attempted to minimise any reporting bias by ensuring a non-treating physician conducted the interviews, avoidance of leading questions, and favouring open-ended questions rather than Linkert scales or questionnaires for evaluating response to treatment. Our treatment outcomes are higher than previously reported with prophylactic medications, perhaps related to the fact that this study was conducted at a tertiary centre, where patient confidence in treating teams may be higher than across non-specialist centres. However, such factors would have applied to both ‘flunarizine’ and ‘other medication’ groups, suggesting these were unlikely to be of direct relevance to the current findings. We would, however, encourage further studies to replicate our findings across different treatment settings.

## 5. Conclusions

There remains a need for good quality randomised control trial data for VM prophylaxis. There is a growing body of evidence supporting the use of flunarizine for VM prophylaxis and we have shown symptomatic benefit in vestibular symptoms in 90% of VM patients taking flunarizine, compared to only 32% with other commonly-used agents. Whilst 50% of the patients reported side-effects with flunarizine, these were not deemed to outweigh the clinical benefits, with most patients deciding to continue treatment. Our data supports the need to include flunarizine in future clinical trials in VM treatment and lends weight to the need to make this medication more readily available for VM patients in the UK.

## Figures and Tables

**Figure 1 brainsci-12-00415-f001:**
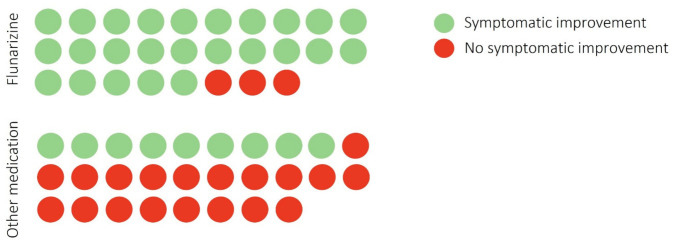
Symptomatic outcome with flunarizine versus ‘other medications’. Each circle represents a patient.

**Figure 2 brainsci-12-00415-f002:**
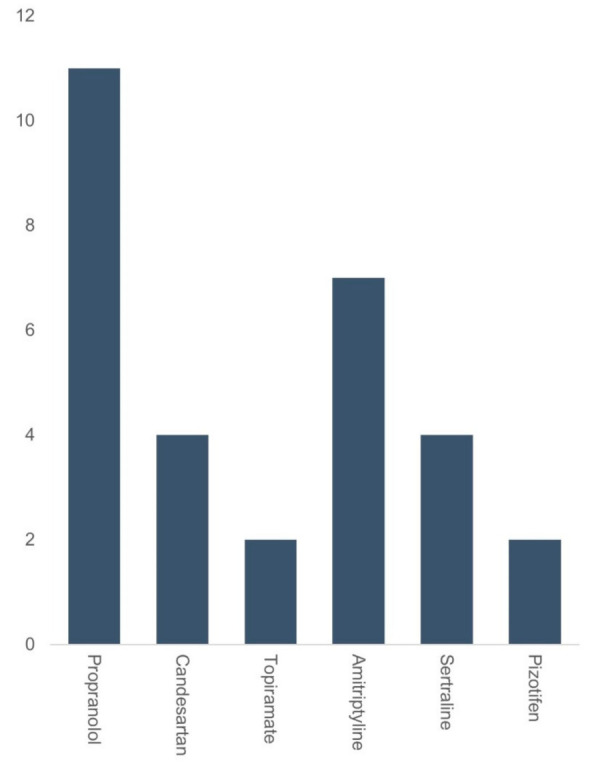
Frequency plot showing the number of patients taking each type of preventative medication, prior to starting flunarizine.

**Table 1 brainsci-12-00415-t001:** Patient clinical and demographic data for Group 1 (Flunarizine) and Group 2 (‘Other medication’).

Groups	Flunarizine	Other Medications	*p*-Value
Mean age in yrs (range)	44.6 (24–70)	50.1 (25–71)	**0.11**
Gender F/M	24/4	23/5	**-**
Disease Duration in yrs (SD)	5.6 (3.5)	4.7 (3.4)	**0.33**
Average monthly vertigo attacks (days)	6.0 (4.0)	5.5 (3.3)	**0.61**

## Data Availability

Data will be provided upon reasonable request to the corresponding author.

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
