# Peer review of "Patient Experience of Flunarizine for Vestibular Migraine: Single Centre Observational Study"

_brainsci, 2022, doi:10.3390/brainsci12040415_

Round 1
Reviewer 1 Report
Very interesting paper with clinical practical use. As a minor revision I'd like to suggest writing the abbreviations meaning before citing them throughout the text.
Author Response
We are grateful to the Reviewer for the positive comments. We have included the definition of any abbreviations in the text before these are cited and apologise for this omission in the earlier version.
Reviewer 2 Report
This article study explores the efficacy and safety of flunarizine in vestibular migraine. It is a retrospective, single-centre observational study. The overall quality is not high, because the study is not randomized nor controlled or blinded. However, I think that this paper may add precious and useful data to the literature, adding new evidence, but I have some concerns:
-abstract: “lines 24-25”. I suggest to replace with a more general statement, for example “Our data supports the use of flunarizine in VM”. The same for the conclusions in lines 235-238, should be rewritten reducing emphasis on these results.
-“ The pathophysiology of vestibular migraine is incompletely understood”. Indeed, it is very debated. I suggest to cite recent studies exploring its implications (Migraine as a Cortical Brain Disorder. Headache 2020). A mention of genetic forms of migraine is needed (Diagnostic and therapeutic aspects of hemiplegic migraine, JNNP 2020). Of interest, FHM2 cases usually presents with vestibular symptoms and can be classified as VM.
-lines 205-208. The evidence provided in the references is still low. In a recent review (Diagnostic and therapeutic aspects of hemiplegic migraine, 2020), an updated evidence is provided, summarizing the current evidence for the therapy in the acute setting and prophylaxis of migraine also providing evidence for mechanisms of efficacy.
-“Neuroimaging studies support the hypothesis that there are spe- 181 cific abnormalities in the structure and activity of the vestibulo-thalamo-cortical pathway 182 in vestibular migraine”. There might be a role for the insular cortex in VM?
- the control group is quite heterogeneous. This might have influenced the results in terms of efficacy and safety, if compared with the group taking flunarizine. This should be accounted as a major limitation of the study. Moreover, the two groups are not balanced in number (28 vs 20).
-“these studies did not directly explore patient experiences with these medications, although one would expect these to correlate with quantitative questionnaire scores of symptom severity. (lines 148-149)”. The author commented on other studies, but I find the same problems in this study… What did the authors define as “symptom improvement or worsening”? It should be clearly stated in the methods and quantified through apposite scales. I suggest to use MIDAS, HIT, or any other objective scale. If these data are not available, it should be clearly discussed as a limitation.
Round 2
Reviewer 2 Report
The authors have answered and modified the manuscript enough. I have no further comments.
This manuscript is a resubmission of an earlier submission. The following is a list of the peer review reports and author responses from that submission.